# Self-Oxygen Regulator System for COVID-19 Patients Based on Body Weight, Respiration Rate, and Blood Saturation

Indrarini Dyah Irawati [1,*], Sugondo Hadiyoso [1], Akhmad Alfaruq [1], Atik Novianti [1] and Achmad Rizal [2]

1   School of Applied Science, Telkom University, Bandung 40257, Indonesia;
    sugondo@telkomuniversity.ac.id (S.H.); alfaruq@outlook.com (A.A.);
    atiknovianti@telkomuniversity.ac.id (A.N.)
2   School of Electrical Engineering, Telkom University, Bandung 40257, Indonesia;
    achmadrizal@telkomuniversity.ac.id
*   Correspondence: indrarini@telkomuniversity.ac.id

**Abstract:** One of the symptoms that appears in patients with COVID-19 is hypoxia or a lack of oxygen in the body's tissues or cells below the proper level. One of the methods used to treat hypoxia is to provide oxygen to the patient. Another device that is needed in oxygen therapy for the patient is an oxygen regulator. An oxygen regulator is needed to regulate the volume of oxygen released to the patient. Currently, the control of oxygen flow by the regulator is still done manually. Therefore, in this study, an oxygen regulator was designed that has the ability to regulate the volume of oxygen output based on body weight, respiration rate, and blood saturation. Using these three parameters, the volume of oxygen to be released is adjusted according to the patient's needs. The system consists of a temperature sensor, mlx90614, and an oxygen saturation sensor, Max30102. The data from the two sensors are processed using microcontrollers to control the movement of the stepper motor as a regulator of the oxygen output volume. The test results show that the system can control the oxygen regulator automatically with a delta error of 0.5–1 L/min. This device is expected to be used for COVID-19 patients who are undergoing self-isolation or who are outpatients.

**Keywords:** COVID-19; automatic oxygen regulator; prototype

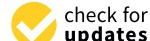

## 1. Introduction

The COVID-19 (Corona Virus Disease) outbreak has spread throughout the world since the end of 2019. This global pandemic has impacted 188 countries globally. The number of positive cases and the number of deaths increased rapidly. Then, the Delta variant appeared and resulted in a quick spike. Delta became the predominant SARS CoV-2 variant with over 99% of COVID-19 cases causing a tremendous increase in hospitalizations in several countries, and more recently a new mutation of the Omicron virus has started to attack. This needs attention because based on experience from the second wave, the capacity of hospitals for COVID-19 patients is running low, so hospitals have had to make decisions that allow confirmed COVID-19 patients with mild symptoms to self-isolate at home [1]. However, with the number of patients who are now self-isolating, there has also been an increase in the indications of breathlessness and respiratory failure in those patients. This is suspected to be due to saturation or oxygen levels suddenly dropping without realizing it.

COVID-19 patients undergoing self-isolation must be aware of the factors causing saturation or blood oxygen (SpO2) levels to drop below normal [2]. The reason is that if the saturation or oxygen levels in the blood drop far below the normal limits, it can cause death. One of the actions for patients with these symptoms is oxygenation of the lungs using pure oxygen through a tube. The medical standard oxygen regulator works manually; there is no automation insofar as the infusion controller is concerned. To regulate the pressure of oxygen coming out of the tube, a regulator is needed. This control is

very important for adjusting to the patient's needs and, of course, for efficiency. Control parameters including respiratory rate and blood oxygen level or oxygen saturation have been widely developed [3–5].

An early identification and monitoring system for routine health checks in IoT-based COVID-19 patients was developed by [6,7]. The system measured heart rate, body temperature, and SpO2 parameters. Motta, L. P. et al. developed a COVID-19 emergency system for the real-time monitoring of oxygen saturation (SpO2), beats per minute (BPM), body temperature (BT), and peak expiratory flow (PEF). It was the first study to evaluate the use of PEF in COVID-19 patients [8]. This home-monitoring device can be controlled independently by the patient [9–11], though it still involves remote control by medical personnel [12–15].

Considering these problems, we propose to create an automatic oxygen regulator system that works based on the respiration rates and oxygen saturation measurements in COVID-19 patients with pneumonia symptoms. In [16], a dynamic model of an electronic oxygen regulator (EOR) was designed for flight using electronic servo control technology. Disturbance-Observer-Based Control (DOBC) with a backstepping method regulates breath pressure stably. In [17], an automatic oxygen regulator system based on respiratory rate and oxygen saturation was designed and realized. However, the oxygen flow control is limited to 5–8 L per minute, though in some conditions a higher oxygen flow is required. In [18], a digital sensor to measure the respiration rate was designed. Our proposed system consists of a breathing sensor, an oxygen saturation sensor, a regulator with modified mechanics, a servo motor as an actuator, and a microcontroller control. This system can simplify settings and provide accurate control according to individual needs and can be used for home care.

## 2. System Design and Implementation

This section discusses the materials used in the system design and then discusses in detail the implementation of the proposed system. In general, this system consists of sensors, processing, input, actuators, and display units as shown in Figure 1. The software design is also discussed in the following subsections.

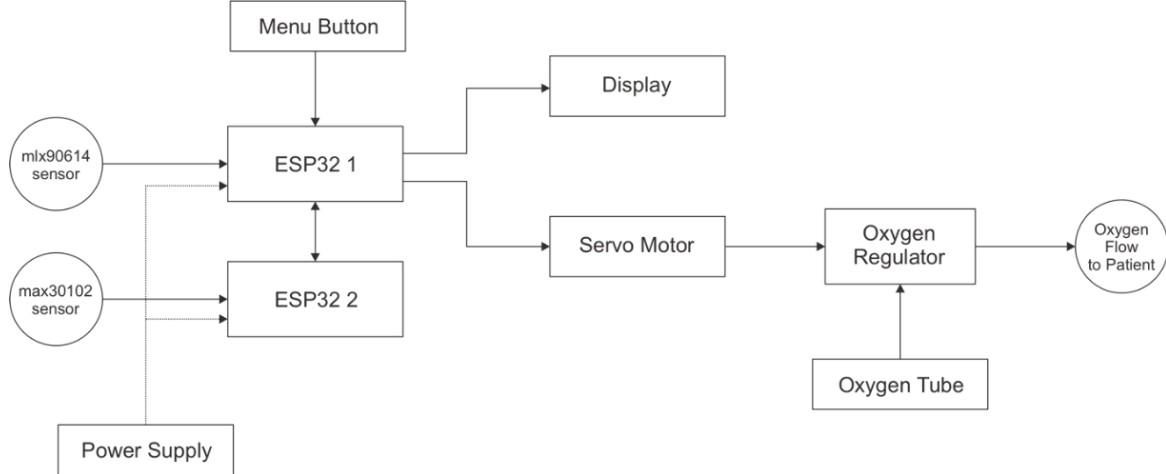

**Figure 1.** Proposed design.

This system uses a temperature sensor mlx90614 to detect the patient's respiration based on changes in temperature during the inspiration and expiration phases. The max30102 sensor calculates oxygen saturation. The servo motor regulates the oxygen output from the regulator. A 3.5 TFT LCD and a push-button act as the display and user interface. Meanwhile, the ESP32 microprocessor is the main controller unit. Due to the need for real-time sensor data retrieval, fast processing is needed and cannot be delayed due to other processes, such as displaying data on a display or driving a motor. Therefore, two microcontrollers, namely ESP32, have a dual CPU that can work independently so

that the processes can be executed simultaneously. The first core in the first ESP32 is used to control the LCD, control the motor, and read the input button. The second core in the first ESP32 is used to read the respiration data. In the second ESP32, both cores read the oxygen saturation sensor. Meanwhile, the second core in the second ESP32 is also used as a backup in case of additional sensors. The proposed device uses two microcontrollers because the signal readings for both the LED and the IR sensor are carried out at the same time to retrieve the oxygen saturation value. Therefore, a simultaneous process is needed with a minimum delay, so that taking the LED/IR signal and calculating the SpO2 is done by a separate processor that it is not affected by other delays, such as temperature readings, display updates, button readings, motor adjustments, and other processes. To connect the main and secondary processes, serial2 is used on the ESP32. The schematic design of this system is presented in Figure 2.

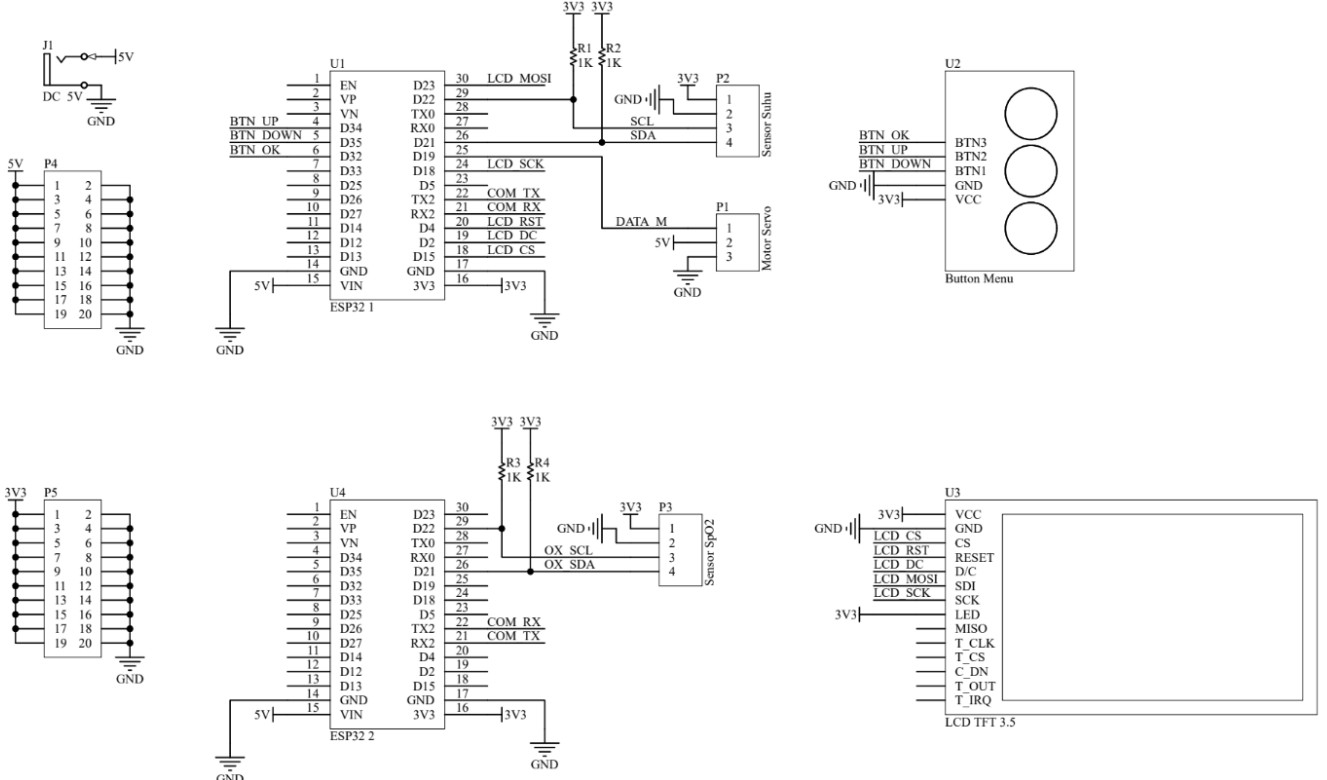

**Figure 2.** Schematic design.

## 2.1. Input and Display

Our design used a 3.5-inch TFT LCD to display the sensor values and oxygen output and a display for the configuration menu, as shown in Figure 3. Serial Peripheral Interface (SPI) communication is used to connect the LCD to the microcontroller with the line connection as in Table 1.

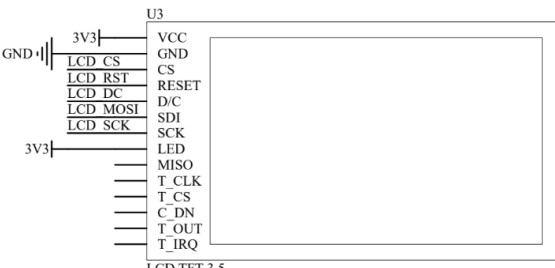

**Figure 3.** TFT LCD schematic.

**Table 1.** Communication between LCD and ESP32 Board 1.

| No | LCD TFT 3.5 | ESP32 Board 1 |
|----|-------------|---------------|
| 1 | VCC | 3V3 |
| 2 | GND | GND |
| 3 | CS | D15 |
| 4 | RESET | D4 |
| 5 | D/C | D2 |
| 6 | SDI | D23 |
| 7 | SCK | D18 |
| 8 | LED | 3V3 |

As shown in Figure 4, three buttons are used: a menu/ok button, an up button, and a down button. The menu/ok button is used to enter the configuration menu and to select 'ok'. The up button moves the selection of the menu/value in an upward direction and the down button moves the selection of the menu/value in a downward direction.

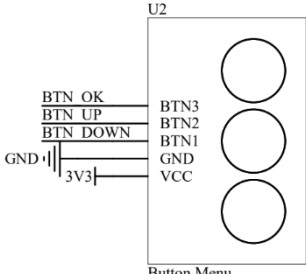

**Figure 4.** Buttons in the system.

The connection between the menu button and ESP32 board is displayed in Table 2.

**Table 2.** Connection between button and ESP32 Board 1.

| No | Menu Button | ESP32 Board 1 |
|----|-------------|---------------|
| 1 | VCC | 3V3 |
| 2 | GND | GND |
| 3 | BTN1 | D15 |
| 4 | BTN2 | D34 |
| 5 | BTN3 | D35 |

*2.2. Oxygen Saturation Measurement*

The oxygen saturation sensor used in this study is the max30102 module. Max 30102 is a high-sensitivity pulse oximeter and heart-rate sensor intended for wearable health. The functional block of max30102 can be seen in Figure 5.

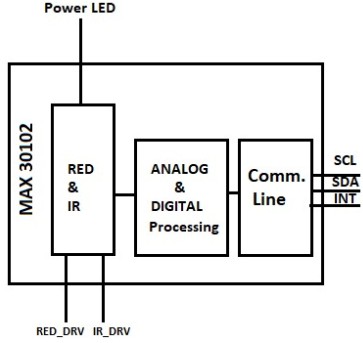

**Figure 5.** MAX30102: Functional Diagram [19].

For communication between the max30102 module and the microcontroller using i2c communication, connecting the SDA and SCL pins of the max30102 module to the SDA and SCL microcontrollers is necessary. Since a long cable between the module and the microcontroller is used, a pull-up resistor is needed on the SDA and SCL pins so that the signal sent is not damaged by noise. Figure 6 displays the max30102 modules connected to the microcontroller.

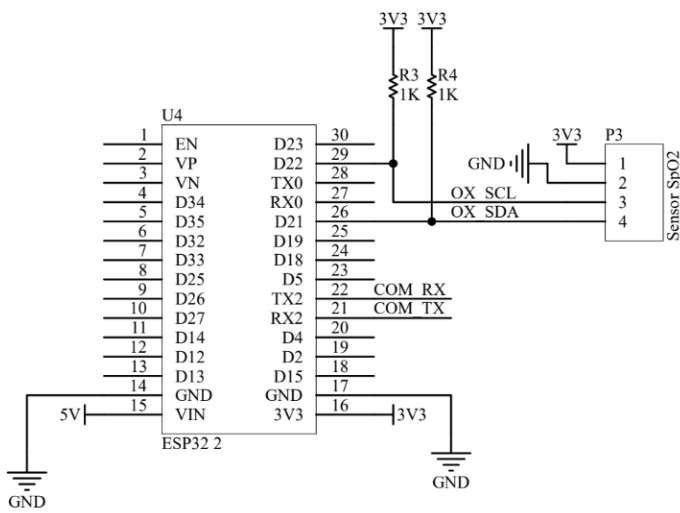

**Figure 6.** Max30102 circuit connected with ESP322.

### 2.3. Respiration Sensor

Each patient's exhalation will result in a change in temperature around the patient's nose/mouth. By utilizing this information, it is possible to increase the rate/rate of respiration by observing changes in temperature around the patient's nose/mouth. We use the temperature sensor, mlx90614to, to capture the temperature changes. Since the communication from mlx90614 uses i2C, it is necessary to connect the SDA and SCL pins of the mlx90614 sensor with the SDA pin and the SCL microcontroller. Figure 7 presents a connection between the temperature sensors with a microcontroller.

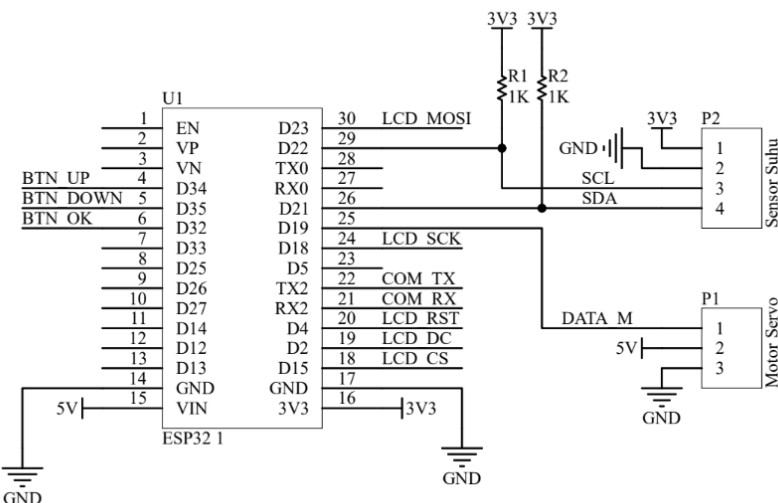

**Figure 7.** Circuit of temperature sensor connected with microcontroller.

### 2.4. Actuator Servo Motor

We use a servo motor to turn the knob on the regulator to adjust the oxygen output from the regulator. A 1:1 gear is used so that the amount of rotation of the regulator knob will be the same as the rotation of the servo motor to connect the servo motor with the

regulator knob. The control pin is connected to pin D19 on the ESP32 board 1 to control the servo motor. The schematic for the servo motor is presented in Figure 8.

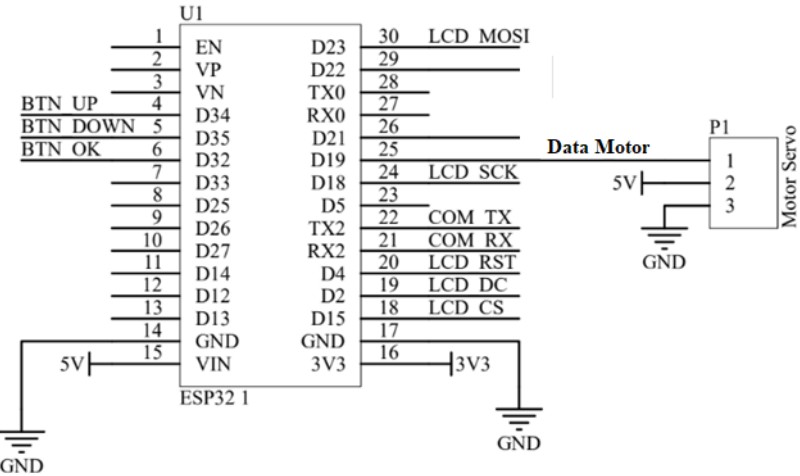

**Figure 8.** Schematic of servo motor.

*2.5. Software Design*

2.5.1. The First ESP32

The first ESP32 core is used to control the flow of the oxygen regulator, display data on the display, read input from the menu button, and read data on serial2. At the same time, the second core is used to read the respiration value from the temperature sensor installed on the mask. When the device is turned on, the microcontroller will activate the first core and the second core to commence the multitasking process. The first core then initializes the LCD and menu buttons; the second core initializes the respiration sensor. After the initialization is successful, it will enter the main menu displaying patient data, the values of the respiration sensor and SpO2, and the amount of oxygen flow to the regulator. In realtime, the first core will always check the data on serial2 to obtain the SpO2 value from the sensor. For example, if the SpO2 value is less than 90, the flow sensor will be calculated according to the formula and then control the servo motor to turn the regulator knob so that the oxygen output matches the formula calculation. In addition to reading the SpO2 value from serial2, the first core also synchronizes with the second core to obtain the patient's respiration value. For example, if the respiration value is greater than 20, the flow sensor will be calculated according to the formula and then control the servo motor to turn the regulator knob so that the oxygen output matches the formula calculation.

A lookup table is made between the amount of oxygen output and the rotation angle of the stepper motor to obtain the oxygen output value from the regulator as desired. The lookup table is presented in Table 3. The block diagram is displayed in Figure 9.

**Table 3.** Lookup table for oxygen output and angle of stepper motor rotation.

| No | Output Oxygen | Angle of Stepper Motor Rotation (Degree) |
|----|--------------|------------------------------------------|
| 1 | 0 | 165 |
| 2 | 1 | 135 |
| 3 | 2 | 120 |
| 4 | 3 | 100 |
| 5 | 4 | 95 |
| 6 | 5 | 85 |
| 7 | 6 | 70 |
| 8 | 7 | 55 |
| 9 | 8 | 40 |
| 10 | 9 | 30 |
| 11 | 10 | 20 |
| 12 | 11 | 10 |

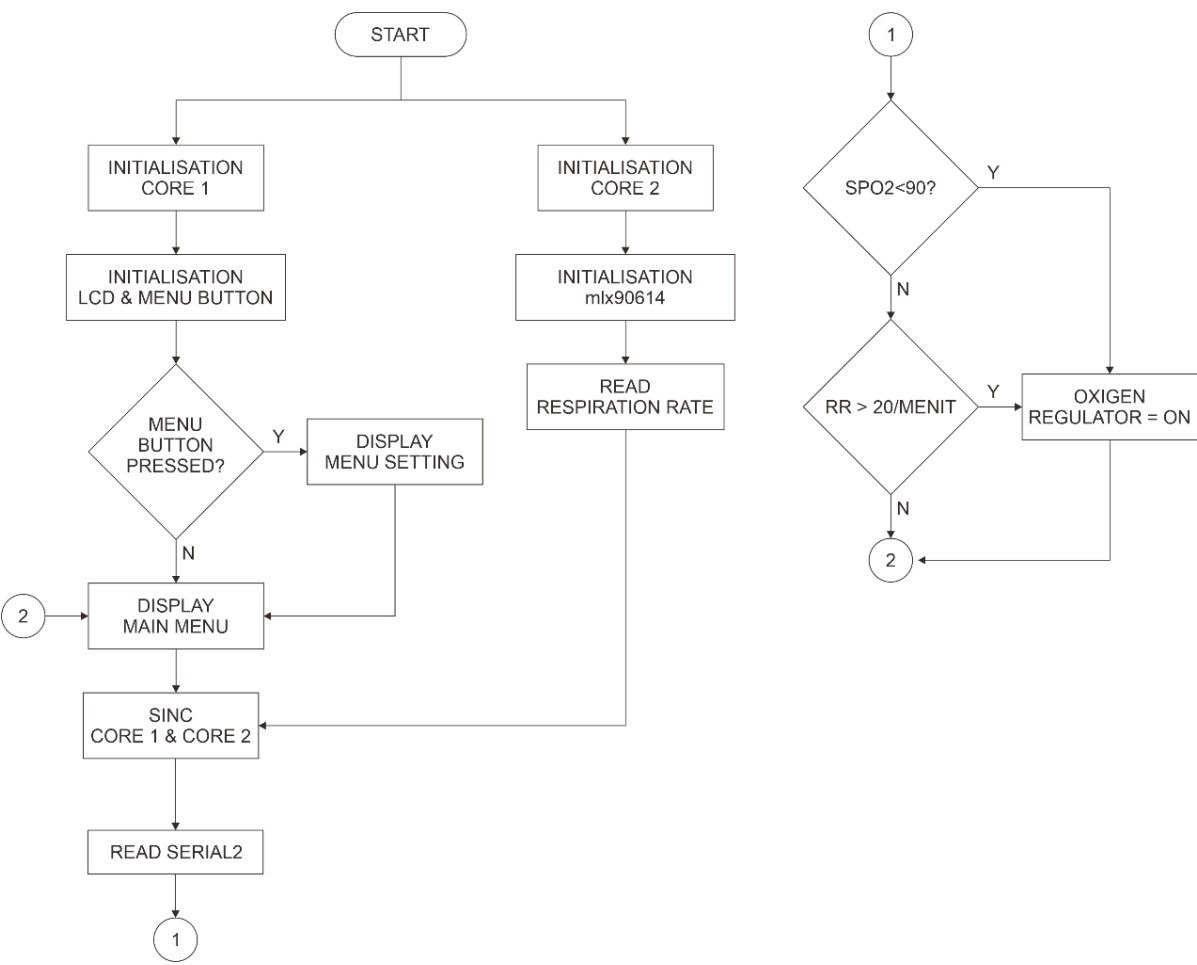

**Figure 9.** Flowchart of setting rotation of motor stepper and oxygen output.

### 2.5.2. Second ESP32

The second ESP32 is used to read the max30102 sensor which is a sensor for reading oxygen saturation, flowchart can be seen Figure 10. When the device is turned on, the microcontroller performs a serial2 initialization and a sensor initialization of max30102. After the initialization is complete, the next step is reading the SpO2 data which is then sent to the second ESP32 via serial2.

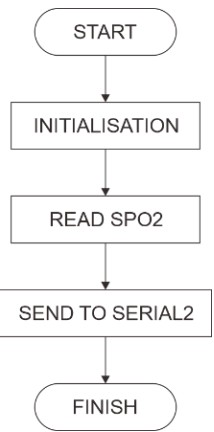

**Figure 10.** Reading process of max30102 by ESP32.

## 3. Results and Discussion

### 3.1. Prototype Realization

In this system, the main controller unit, display unit, and several input interfaces are placed in a box with polylactic acid (PLA) material to avoid short-circuit effects. Boxes are custom-designed using 3D printing. The sensor interface and the regulator control actuator are placed on the back using an 8-pin GX20 socket. The finger clip for the oximeter sensor is also fabricated using 3D printing. The breath sensor is placed on the inside of the mask. Meanwhile, regulators from commercial products are modified in the control flow meter section by adding a stepper motor and several components that are self-fabricated. The proposed product prototype that is realized in this study is presented in Figures 11–13.

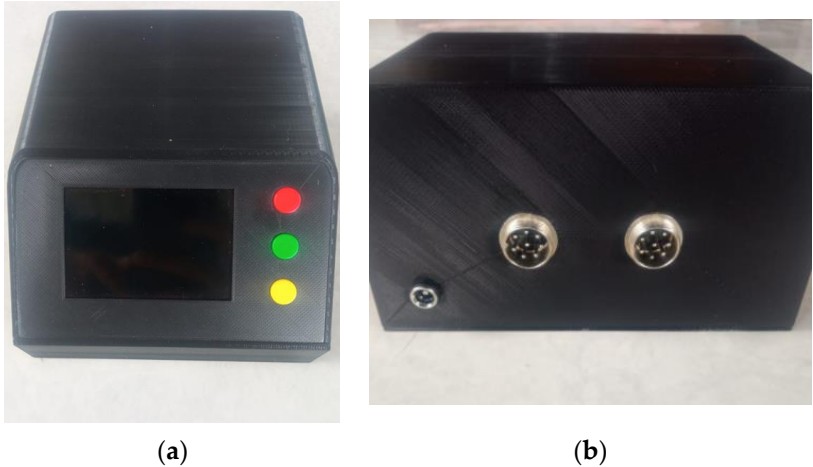

(**a**)　　　　　　　　　　　　　　　(**b**)

**Figure 11.** Box hardware (**a**) front view and (**b**) back view.

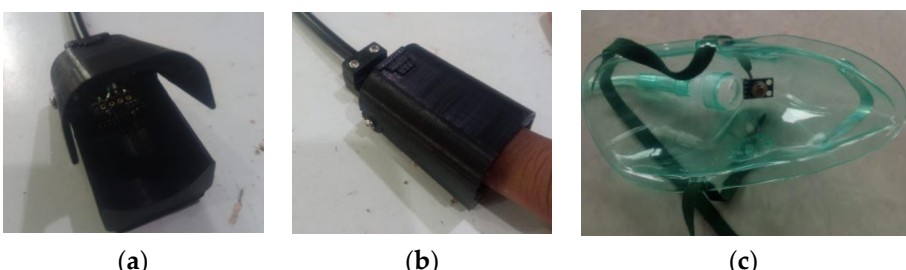

(**a**)　　　　　　　　　(**b**)　　　　　　　　　(**c**)

**Figure 12.** Sensor placement (**a**) in finger clip; (**b**) finger placement in clip; (**c**) respiration sensor.

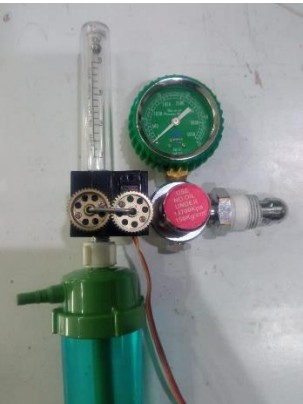

**Figure 13.** Mechanical implementation in regulator.

The sensor and actuator interface with the main controller unit is designed to be compact and plug and play using a fiber cable with high flexibility through the GX20 8-pin socket as the connector. Figure 14 shows the overall device integration.

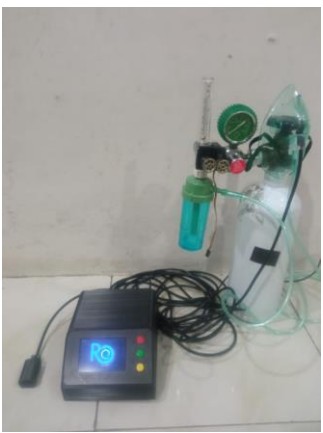

**Figure 14.** System integration.

### 3.2. Sensor Testing

This section discusses in detail the testing of the respiratory sensor and oximeter sensor. This test involved healthy subjects, both adults and children. For the adults, sessions were conducted on how to hold their breath for a certain period. Holding one's breath is expected to result in an oxygen saturation value below 95% and is used to test the accuracy of the sensor at more varied oxygen saturation levels. The test results for each sensor are described in the following sub-sections.

### 3.2.1. Testing of Respiration Rate Sensor

In the breathing sensor testing, an oxygen mask is placed over the mouth and nose. The subject is relaxed and sitting in a chair. The subjects were asked to breathe normally; there was no emphasis on either the inspiratory or expiratory phases. The MLX90614 sensor will detect changes in temperature flowing in the oxygen mask due to the inspiration and expiration phases as shown in Figure 15. The expiratory phase shows a higher temperature than the inspiratory phase. One respiratory cycle includes these two phases which are then estimated every 15 s to obtain the respiration rate per minute. Since the raw signal from the sensor contains a small number of ripples with a high frequency as shown in Figure 16, a digital low-pass filter with a cut-off frequency of 10 Hz is designed to reject the ripples. The filtered signal is shown in Figure 16. A red line indicates that the respiratory signal is not contaminated with high-frequency noise.

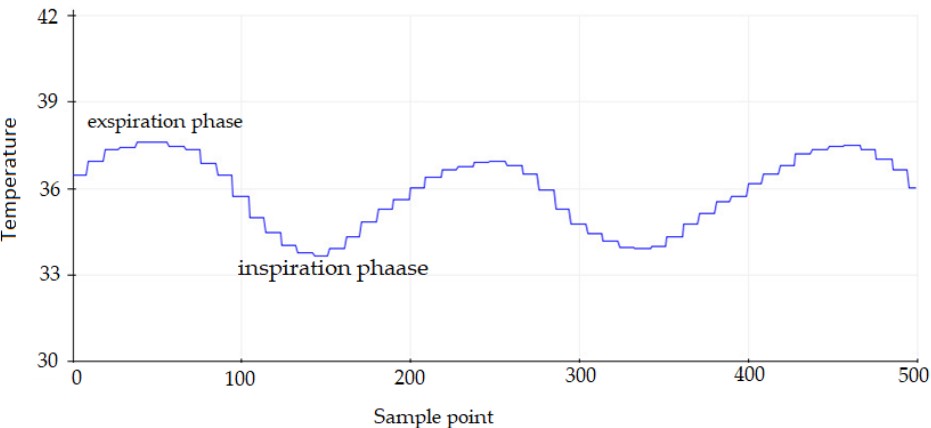

**Figure 15.** Temperature flowing in the mask.

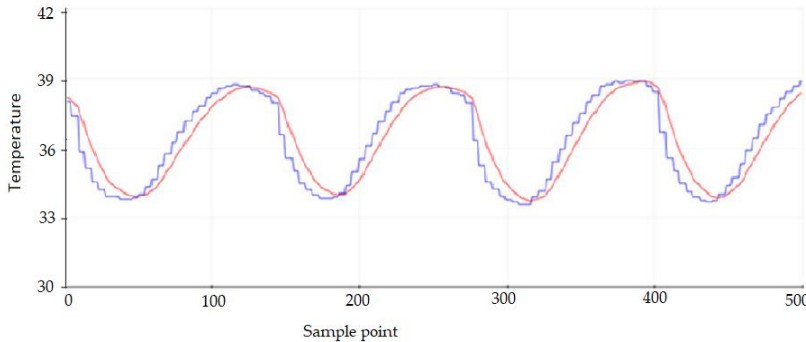

**Figure 16.** High-frequency ripple and filtered signal.

The results of the calculation of the respiratory rate by the main controller unit based on sensor readings are then compared with the manual calculations with palpation techniques for one minute. The tests for adults and children were carried out 15 times each. The test results are presented in Table 4. The test results show that the calculation of the respiratory rate using the sensor produces an accuracy of up to 100% with a delta error of ±1–2 rate/min both over and under estimation. This delta error could be caused by the calculation of breathing by the system for 15 s, which is then multiplied by four to obtain the respiratory rate per minute. Thus, the respiratory rate estimated by the device is a multiple of four. Therefore, it is possible to generate a delta error compared to the actual respiratory rate.

**Table 4.** Comparative results of respiration rate calculation on normal breathing.

| Subject | Measurement No. | RR Sensor | RR Manual | Delta |
|---|---|---|---|---|
| Adult | 1 | 12 | 12 | 0 |
| | 2 | 8 | 8 | 0 |
| | 3 | 8 | 9 | 1 |
| | 4 | 8 | 9 | 1 |
| | 5 | 16 | 15 | 1 |
| | 6 | 12 | 10 | 2 |
| | 7 | 12 | 12 | 0 |
| | 8 | 12 | 12 | 0 |
| | 9 | 12 | 11 | 1 |
| | 10 | 8 | 8 | 0 |
| | 11 | 16 | 16 | 0 |
| | 12 | 20 | 22 | 2 |
| | 13 | 8 | 8 | 0 |
| | 14 | 8 | 8 | 0 |
| | 15 | 8 | 8 | 0 |
| Children | 1 | 24 | 24 | 0 |
| | 2 | 24 | 24 | 0 |
| | 3 | 24 | 24 | 0 |
| | 4 | 28 | 27 | 1 |
| | 5 | 28 | 26 | 2 |
| | 6 | 24 | 24 | 0 |
| | 7 | 24 | 24 | 0 |
| | 8 | 20 | 22 | 2 |
| | 9 | 24 | 24 | 0 |
| | 10 | 24 | 25 | 1 |
| | 11 | 24 | 24 | 0 |
| | 12 | 28 | 28 | 0 |
| | 13 | 28 | 28 | 0 |
| | 14 | 24 | 24 | 0 |
| | 15 | 20 | 20 | 0 |
| | Delta | | | ±1–2 rate/min |

To test the reliability of the performance of the breathing sensor, another test scenario was also carried out with very fast breathing in the range of 30–70 breaths per minute. Adult subjects were asked to breathe faster as if experiencing shortness of breath. The results of testing this scenario are presented in Table 5. In this scenario, the device is able to achieve 100% accuracy even though in some experiments there is a delta error of $\pm$2–6 rate/min. In the respiratory scenario greater than 30 rate/min., the resulting delta error also tends to be linear.

**Table 5.** Fast-breathing measurements.

| Measurement No. | RR Sensor | RR Manual | Delta |
|:---:|:---:|:---:|:---:|
| 1 | 48 | 52 | 4 |
| 2 | 48 | 53 | 5 |
| 3 | 52 | 54 | 2 |
| 4 | 52 | 52 | 0 |
| 5 | 48 | 45 | 3 |
| 6 | 60 | 60 | 0 |
| 7 | 56 | 56 | 0 |
| 8 | 48 | 48 | 0 |
| 9 | 36 | 36 | 0 |
| 10 | 36 | 36 | 0 |
| 11 | 76 | 70 | 6 |
| 12 | 48 | 48 | 0 |
| 13 | 36 | 36 | 0 |
| 14 | 48 | 44 | 4 |
| 15 | 52 | 48 | 4 |
| | Delta | | $\pm$2–6 rate/min |

### 3.2.2. Testing of the Oxygen Saturation Sensor

Testing the performance of the oxygen saturation sensor was carried out on adult and pediatric subjects. The measurement results are then compared with the commercial product oximeter "Mixio". To minimize reading errors, the sensor is placed on the forefinger and the Mixio is placed on the middle finger of the same arm. This test was carried out 30 times, and for several measurements the adult subjects were asked to hold their breaths. The measurement results of the oxygen saturation sensor are presented in Table 6. From this test, it is known that the max30102 sensor is able to read oxygen saturation with a delta error of $\pm$1–2%, where the error value is still within the limits of medical recommendations. This difference in the readings by the sensor and the Mixio could be caused by finger movements that may occur spontaneously. With sensor readings that meet the recommendations, this sensor can be used to reliably measure the oxygen saturation in this proposed system.

**Table 6.** Oxygen saturation measurement.

| Measurement No. | SpO2 (%) Sensor | SpO2 (%) Mixio | Delta | Measurement No. | SpO2 (%) Sensor | SpO2 (%) Mixio | Delta |
|:---:|:---:|:---:|:---:|:---:|:---:|:---:|:---:|
| 1 | 99 | 99 | 0 | 16 | 96 | 96 | 0 |
| 2 | 98 | 98 | 0 | 17 | 99 | 99 | 0 |
| 3 | 99 | 99 | 0 | 18 | 98 | 98 | 0 |
| 4 | 98 | 98 | 0 | 19 | 97 | 97 | 0 |
| 5 | 97 | 97 | 0 | 20 | 97 | 96 | 1 |
| 6 | 98 | 97 | 1 | 21 | 97 | 98 | 1 |
| 7 | 96 | 97 | 1 | 22 | 99 | 98 | 1 |
| 8 | 95 | 95 | 0 | 23 | 98 | 99 | 1 |
| 9 | 93 | 94 | 1 | 24 | 98 | 99 | 1 |
| 10 | 90 | 92 | 2 | 25 | 98 | 98 | 0 |
| 11 | 88 | 88 | 0 | 26 | 99 | 99 | 1 |
| 12 | 87 | 88 | 1 | 27 | 95 | 96 | 1 |
| 13 | 88 | 88 | 1 | 28 | 95 | 95 | 0 |
| 14 | 86 | 86 | 0 | 29 | 96 | 96 | 0 |
| 15 | 85 | 86 | 1 | 30 | 98 | 98 | 0 |
| | Delta | | | | | | $\pm$1–2% |

### 3.2.3. Testing of the Automatic Regulator

The proposed system supports both semi-automatic and fully-automatic control applications. Semi-automatic means that the user can set the MV manually through the device interface and the processor will then send commands to the actuator to adjust the oxygen rate through the servo motor. In this mode, the control of the regulator does not refer to body weight, respiration rate, or oxygen saturation. In fully-automatic mode, the system will adjust the oxygen rate in the regulator based on the body weight, respiration rate, and oxygen saturation data that are read by the sensor. Tests of the performance of the system in controlling the oxygen rate were carried out in fully-automatic mode, which also represents the test in semi-automatic mode.

This performance test only involved adult subjects. Subjects were asked to breathe at various rates, which was necessary to test the response of the actuator when receiving a low-to-high flow-control instruction or vice versa. The variations in the weight values are also set manually to make testing easier. The respiration rate is conditioned to greater than 20 rate/min, which is an abnormal condition for adults, so that the automatic mode works. There are four variations of respiration rate for each weight value.

The test results of the proposed system in automatic mode are presented in Table 7. The results of the MV calculation by the device are rounded up/down so there are differences from the manual calculations. This is intended to simplify the step of regulating the oxygen flow, but with small deviations of about 0.5–1 L/min.

The test results of the variation of the oxygen flow value from high to low and vice versa do not affect the performance of the actuator. In other words, the actuator remains accurate in responding to changes in the oxygen flow according to the commands given by the main control unit.

A performance evaluation was also carried out by comparing the performance of the proposed system with a previous study by Puspitasari et al. [17]. In the previous study, the maximum controllable oxygen flow was 8 L/min. Meanwhile, the currently proposed device is capable of delivering oxygen up to 14 L/min. In some conditions, for example, patients with a higher body weight and respiratory rate, more oxygen supply is required.

**Table 7.** The test results of the proposed system in the control of the regulator in automatic mode.

| Body Weight (Kg) | Resp. Rate (RR/min.) | Min. Vent. (L/min.) | Proposed Device | Regulator Flow Meter | Delta Flow |
|---|---|---|---|---|---|
| 60 | 20 | 7.2 | 7 | 7 | 0 |
|  | 20 | 7.2 | 7 | 7 | 0 |
|  | 28 | 10.08 | 10 | 10 | 0 |
|  | 24 | 8.64 | 9 | 9.5 | 0.5 |
| 70 | 20 | 8.4 | 8 | 8 | 0 |
|  | 24 | 10.08 | 10 | 11 | 1 |
|  | 28 | 11.76 | 12 | 12.5 | 0.5 |
|  | 16 | 6.72 | 7 | 7 | 0 |
| 50 | 24 | 7.2 | 7 | 7 | 0 |
|  | 28 | 8.4 | 8 | 8 | 0 |
|  | 48 | 14.4 | 14 | 14 | 0 |
|  | 32 | 9.6 | 10 | 10 | 0 |
| 55 | 16 | 5.28 | 5 | 5.5 | 0.5 |
|  | 28 | 9.24 | 9 | 9 | 0 |
|  | 32 | 10.56 | 11 | 11 | 0 |
|  | 24 | 7.92 | 8 | 8 | 0 |
| 65 | 28 | 10.92 | 11 | 12 | 1 |
|  | 20 | 7.8 | 8 | 8 | 0 |
|  | 24 | 9.36 | 9 | 10 | 1 |
|  | 36 | 14.04 | 14 | 14 | 0 |

This proposed system can be used by patients who require continuous oxygen therapy during the medical treatment period. This system is able to control the flow of oxygen based on the needs of the human body with reference to body weight, oxygen saturation, and respiration rate. This system is expected to provide efficiency in the use of oxygen during the COVID-19 pandemic.

## 4. Conclusions

A prototype of an automatic oxygen regulator (ROO) for COVID-19 patients has been successfully created. The ROO works based on body weight, respiration rate, and the oxygen saturation in the blood. The system was tested on 30 subjects consisting of 15 adults and 15 children. The respiration rate sensor produces 100% accuracy with a delta error of $\pm1$–2 rate/min both over and under estimation. In testing the oxygen saturation in the blood, the measurement error is $\pm1$–2%. Overall system testing shows that the automation in oxygen rate regulation has small deviations of about 0.5–1 L/min. In the initial testing phase, this prototype was only tested for home use. All the subjects agreed to complete the tests. The prototype is guaranteed to be safe as the installed sensors do not pose an electrical hazard. For further implementation, it is necessary to conduct clinical trials on patients in hospitals. Another solution would be to make a prototype tool that is smaller in size so that it is portable and at the same time monitoring can be done remotely.

## 5. Patents

This research resulted in the intellectual property of the brand for the automatic oxygen regulator, namely ROO.

**Author Contributions:** Conceptualization, I.D.I. and S.H.; methodology, S.H. and A.A.; hardware, A.A.; validation, S.H; formal analysis, I.D.I. and S.H.; investigation, I.D.I., S.H., A.A., A.N. and A.R.; resources, I.D.I., S.H., A.A., A.N. and A.R.; data curation, A.A. and S.H.; writing-original draft preparation, I.D.I., S.H. and A.R.; writing-review and editing, A.N; visualization, A.R. All authors have read and agreed to the published version of the manuscript.

**Funding:** This research was supported by Telkom University based on the decision of the Vice Rector KWR4/032/PNLT2/PPM-LIT/2021. Financing for the program "Penelitian Pengembangan Unggulan dan Inovasi Teknologi Produk (PPU-PTI)" in 2021 for "Regulator Oksigen Otomatis (ROO)" project.

**Informed Consent Statement:** Informed consent was obtained from all subjects involved in the study.

**Acknowledgments:** This work is supported by the Laboratories of Diploma Telecommunication Engineering and Telkom University.

**Conflicts of Interest:** The authors declare no conflict of interest.

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
