# Peer review of "Self-Oxygen Regulator System for COVID-19 Patients Based on Body Weight, Respiration Rate, and Blood Saturation"

_electronics, doi:10.3390/electronics11091380_

Round 1
Reviewer 1 Report
The study by Rizal et al, reports on a system that can regulate the oxygen in Covid-19 patients. The oxygen released is regulated/adjusted based on body weight, respiration rate, and blood saturation. The system consists of a temperature sensor and an oxygen saturation sensor. The data from the two sensors inform the regulator of the oxygen output volume.
I do not have any specific concerns regarding the results or the experimental design of the current work. However this work is highly technical and mainly involves testing of a prototype device. The device itself in terms of engineering and design (i.e., the sensors used or the way they are combined) does not involve any novelty to the best of my knowledge. The resulting device may be of interest from a commercial/ practical standpoint but from a scientific publication standpoint I believe it should be presented in a highly technical journal.
Author Response
Thanks for the suggestions
Indeed, this is an applied research funded due to the covid1-9 pandemic, but it is still in the scope of issue of the journal suggested by the editor of Recent Advanced Applications of Rehabilitation and Medical Robotics.
I have put all the suggestions on the paper in the highlight sentence. Thank you.

Reviewer 2 Report
We thank the authors for the effort provided to propose an oxygen self-regulation system for the Covid-19 patient. The system consists of a temperature sensor and an oxygen saturation sensor. The data from the two sensors is processed using two microcontrollers to control the stepper motor movement as oxygen output volume regulator.
However, from a presentation and editorial perspective, the introduction is very poor in terms of literature review. Many figures do not have added values to put in the text such as figures 2, 3, 4, 5, 6 and 7.
From a technical point of view, I am not convinced by the proposed design. Why use two microcontrollers to control the oxygen regulator. I'm not convinced by the multitasking and real-time aspect of your application. Anyway, we are not in this scale of real time constraints. The application is just for acquiring temperature and oxygen saturation and based on this data you control a seromotor. As you may have noticed, you may have wasted some time in the serial communication between the two microcontrollers. It is therefore an additional time that can be avoided if a single microcontroller is used.
Author Response
Thank you for your suggestions.
All images have the aim of making it easier for readers to understand because it is related to the design of a medical device system if you want to implement it.
Why use two micro controllers to control the oxygen regulator? It is necessary to read the RED and IR sensor values at the same time to get the SPO2 value. So that this value is not delayed when taking RED and IR samples, a simultaneous process is needed with a minimum delay. Therefore, RED and IR sampling and SPO2 calculations are carried out on separate processors so that they are not affected by other delays, such as temperature readings, display updates, button reading, motor adjustment, etc.
I have put all the suggestions on the paper in the highlight sentence. Thank you very much.

Reviewer 3 Report
1.359 / 5.000Rezultatele traducerii
The paper presents a topical issue, namely the development of a prototype automatic oxygen regulator for patients with covid-19. The abstract is very good, it briefly presents what the work approaches. The introductory part is well done with recent references. I especially appreciate the structure of the work, which has a fair organization. The second part describes the system and how to implement it. In the 3rd part called Results and discussions, the realization of the prototope is presented as well as the experimental test part. In this part, information is collected from the saturated oxygen saturation sensor, the number of breaths per minute, as well as the automatic regulator. As suggestions for improving the work I would have: Figures 15, 16 can be improved because the values ​​written on the axes are too small to be read. In the conclusions, the authors say that this system has been tested on a number of 30 patients. I would have expected that in future research the authors would foresee its use in hospitals and not its minimization. Another issue that needs to be clarified is whether this prototype is subject to approval for use by patients in hospitals. I appreciate the scientific contribution brought by this work and I congratulate the authors for their idea and achievement.Author Response
Thank you for your appreciations.
We've fixed the values written on the axes that can be read in figure 15 and 16. At the initial implementation stage, testing was limited to patients at home and had received approval to be involved in the study. And sensors installed from the safety side do not pose an electrical hazard. For further implementation, it needs to be clinically tested on patients at the hospital. And has been included in the research suggestion.
All of your suggestion have put in our paper and you can check it on the highlight sentences. Thank you very much.

Reviewer 4 Report
One of the symptoms that appear in patients with Covid-19 is hypoxia or lack of oxygen in the body's tissues or cells below the proper level. One of the methods used to treat hypoxia is to provide oxygen to the patient. Devices other than oxygen cylinders needed to provide oxygen to patients are oxygen regulators. An oxygen regulator is needed to regulate the volume of oxygen released to the patient.
In this study, the authors designed an oxygen regulator that has the ability to regulate the volume of oxygen output based on body weight, respiration rate, and blood saturation.
Using these three parameters, the volume of oxygen to be released is adjusted according to the patient's needs. The system consists of a temperature sensor mlx90614 and an oxygen saturation sensor Max30102. The data from the two sensors is processed using two microcontrollers to control the movement of the stepper motor as a regulator of the oxygen output volume.
They successfully tested the system in the required clinical ranges of application
The contribute is very interesting.
The system is really well described and really useful.
As a biomedical sensor designer I appreciated it very much.
Some minor suggestions with an academic spirit:
- Better balance the abstract with the summary of the sections. The summary of the conclusions is for example lacking
- You put in the introduction this text as aim “Our proposed system consists of a breathing sensor, oxygen saturation sensor, a regulator with modified mechanics, a servo motor as an actuator, and a microcontroller control. This system can simplify settings and provide accurate control according to needs and be used for home care.” I’d enlarge it and I’d put it in a paragraph dedicated to the purpose.
- Very good the schematic entry of the system in the second paragraph dedicated to the system design and implementation. You inserted the final product and its test in a paragraph named “results and discussion”. Usually the discussion is dedicated to the comparison with other studies, to the description of the limits and other aspects. Therefore a true discussion is lacking.
Author Response
Thank you for you suggestions.
- Better balance the abstract with the summary of the sections. The summary of the conclusions is for example lacking,
"We've added the conclusion to a more complete paragraph."
2. You put in the introduction this text as aim “Our proposed system consists of a breathing sensor, oxygen saturation sensor, a regulator with modified mechanics, a servo motor as an actuator, and a microcontroller control. This system can simplify settings and provide accurate control according to needs and be used for home care.” I’d enlarge it and I’d put it in a paragraph dedicated to the purpose.
"We are open to discussion to develop joint research.
3. Very good the schematic entry of the system in the second paragraph dedicated to the system design and implementation. You inserted the final product and its test in a paragraph named “results and discussion”. Usually the discussion is dedicated to the comparison with other studies, to the description of the limits and other aspects. Therefore a true discussion is lacking.
"We have added a comparative discussion with previous studies."
All of revision have put on highlights sentences.

Round 2
Reviewer 2 Report
There is no change following my remarks given at the first round.
From point of view presentation and writing the paper is not yet ready to be published in an impacted journal. The introduction is very poor in term of litterature review. Many figures are unsuful and doesnt make sense to put them in a scientific paper. Overall, the paper looks like a final year project technical report and it s away to be a scientific paper. From technical point of view, I m still not convinced by this design. Why using two microcontroleurs to control the oxygen regulator. I m not convicend by the multitasking processing and the real time aspect in your application. For a specialist, it is very clear that your equipment is not capable of running a real-time application. Anyway, we are not in this scale of real time constraints. The application is just to acquire temperature and oxygene saturation and based on these data, you control a very basic sero motor. As you noticed perhaps that you have a waste time in serial communication between the two microcontrollers. Therefore it is an additional time that we can avoid if we use a signle microcontroller.
As conclusion, really I m not convinced by this work neither the development nor the presentation. Authors insist on figures presenting basic electronic components datasheets, even in FYP reports we pesent these data in appendix. Actually the presentation of the paper and the formatting are far from being worthy of a scientific paper.
Author Response
Thank you for your valuable comments. We apologize if the previous response did not satisfy you. We have tried to find the literature most relevant to this proposed system. To be honest, the literature reviews that the proposed oxygen regulator automatic is very limited. To the best of our knowledge, the similar paper is only in reference to [17 Puspitasari,A. J.; Famella, D.; Ridwan, M. S.; Khoiri, M. Design of low-flow oxygen monitor and control system for respiration and SpO2 rates optimization. IOP Conf. Series: Journal of Physics: Conf. Series 1436 (2020) 012042. https://doi.org/10.1088/1742-6596/1436/1/012042]. But this paper has weakness that the amount of oxygen delivered is small. Because of it, we propose adjusting the volume of oxygen to suit the needs and can deliver oxygen with a larger volume. For the issue of why use two micro controllers because for oxygen saturation readings it is necessary to read the RED and IR sensor values at the same time.
So that this value is not delayed when taking RED and IR samples, a simultaneous process worked with a minimum delay. Therefore, RED and IR sampling and SPO2 calculations are carried out on separate processors so that they are not affected by other delays, such as temperature readings, display updates, button reading, motor adjustment, etc.
Honestly for the current device we have not tried to use a single microcontroller. In the next development we will try to use a microcontroller with a larger memory and a larger number of I/O so that it can handle all I/O needs and communication lines with sensors. Thank you again for your comments for the development of our proposed device so that it can be more reliable.
